

# Noise-induced transport in the Aubry-André-Harper model

Devendra Singh Bhakuni[1], Talía L. M. Lezama[2] and Yevgeny Bar Lev[1]

**1** Department of Physics, Ben-Gurion University of the Negev, Beer-Sheva 84105, Israel
**2** Department of Physics, Yeshiva University, New York, New York 10016, USA

## Abstract

We study quantum transport in a quasiperiodic Aubry-André-Harper (AAH) model induced by the coupling of the system to a Markovian heat bath. We find that coupling the heat bath locally does not affect transport in the delocalized and critical phases, while it induces logarithmic transport in the localized phase. Increasing the number of coupled sites at the central region introduces a transient diffusive regime, which crosses over to logarithmic transport in the localized phase and in the delocalized regime to ballistic transport. On the other hand, when the heat bath is coupled to equally spaced sites of the system, we observe a crossover from ballistic and logarithmic transport to diffusion in the delocalized and localized regimes, respectively. We propose a classical master equation, which in the localized phase, captures our numerical observations for both coupling configurations on a qualitative level and for some parameters, even on a quantitative level. Using the classical picture, we show that the crossover to diffusion occurs at a time that increases exponentially with the spacing between the coupled sites, and the resulting diffusion constant decreases exponentially with the spacing.

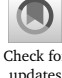
# 1   Introduction

Non-equilibrium dynamics of quantum systems has been a topic of great interest in condensed matter and statistical physics, particularly concerning the study of quantum transport [1–8]. While transport in generic many-body systems is typically diffusive, diffusion of a quantum particle can be suppressed by a random disordered potential due to localization of the single-particle states. This phenomenon, dubbed Anderson localization [9], occurs for any non-zero random disorder in one and two dimensions. At higher dimensions, *all* states are localized only for sufficiently high disorder. For disorder values smaller than a critical value, only eigenstates below a specific energy, called the mobility edge, are localized [10–12].

Localization also occurs for systems with non-random, quasi-periodic potentials. Such potentials can exhibit localization to delocalization transitions and mobility edges, even in one dimension [13–20], and are experimentally feasible [21–24]. In the quasi-periodic Aubry-André-Harper (AAH) potential, for example, all states are localized above a critical potential strength and delocalized otherwise, whereas the transition point features fractal states [13,14]. Transport is absent in the localized phase, ballistic in the delocalized phase, and anomalous at the critical point due to the fractal structure of the eigenstates [25–28]. Localization is primarily restricted to isolated set-ups. Realistic physical systems are inevitably coupled to an environment, such as, phonons, which destroy localization and induce finite DC conductivity [29]. Therefore, it is crucial to understand the impact of the environment on quantum transport.

Anderson localization is known to be stable to local and global periodic driving [8], as well as local and quasi-periodic in time drives [8,30]. However, for a random, uncorrelated time-dependent drive coupled to *all* lattice sites, Anderson localization is destroyed, and diffusive transport is recovered [31–33]. For correlated noise, a transient sub-diffusive regime can emerge [34]. On the other hand, randomly driving a *single-site* of the system induces logarithmic transport [35]. A natural question, that we consider in this work, is how transport is affected when only part of the sites of the lattice are coupled to a random drive.

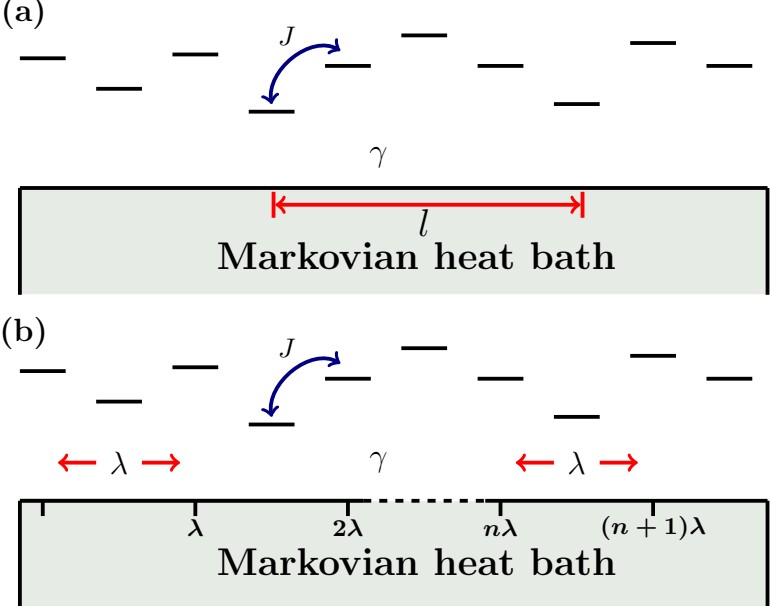

Figure 1: Schematic of the coupling of the heat bath to the system. (a) central region of $l$ sites is coupled (b) sites at $\lambda$ distance apart are coupled.

For a finite density of coupled sites, randomly distributed in the system, a transition from a super-diffusive to diffusive stationary current as a function of the density of the coupled sites was established in Refs. [36, 37]. These works did not consider temporal dependence of spreading excitations, which may have distinct behavior from the behavior of the stationary current [25, 26]. It is, therefore, an open question if the spreading of density excitations exhibits a single or several regimes of transport as a function of time.

Sparse coupling to a heat bath serve as useful phenomenological models to understand the effect of the rare ergodic bubbles [35–42] in many-body localized systems. These bubbles act as a bath, and serve as a destabilizing mechanism of many-body localization [39, 40, 43–47]. While for disordered many-body localized models the ergodic bubbles are randomly distributed, this is not true in quasi-periodic systems. Consequently, exploring how different coupling configurations affect quantum transport in such scenarios is essential. Some of these configurations have been previously studied in experiments and theory but without a direct connection to quantum transport [39, 47].

In this work, we answer the questions above by studying the temporal spreading of density excitations at infinite temperatures for different couplings of the system to a Markovian bath.

The paper is organized as follows. We describe the model Hamiltonian and the methods in Section 2. In Section 3 we present our results for different couplings of the model to a Markovian heat bath. Finally, we discuss our findings in the Section 4.

## 2 Model and methods

We consider a system of spinless fermions in a chain of length $L$, which is described by the Hamiltonian,

$$\hat{H} = -J \sum_{m=1}^{L-1} \left( \hat{a}_{m+1}^{\dagger} \hat{a}_m + \hat{a}_m^{\dagger} \hat{a}_{m+1} \right) + W \sum_{m=1}^{L} \cos\left(2\pi\beta m + \phi\right) \hat{n}_m, \tag{1}$$

where $\hat{n}_m = \hat{a}_m^{\dagger} \hat{a}_m$, and $\hat{a}_m, \hat{a}_m^{\dagger}$ are annihilation and creation operators of a fermion on site $m$, $J$ is the hopping strength, $W$ is strength of the potential and $\beta = \left(\sqrt{5}-1\right)/2$ is the Golden mean. The phases are taken uniformly from $\phi \in [-\pi, \pi]$. In contrast to the Anderson model, where all the single-particle eigenstates are localized for any non-zero value of $W$, the AAH model exhibits a delocalization-localization transition. For $W < 2J$, *all* the single-particle eigenstates are extended and transport is ballistic [26], while for $W > 2J$, *all* the states are localized. At the critical point, $W = 2J$, the states are multi-fractal and transport is diffusive if characterized by the mean squared displacement [26], while a characterization based on the stationary current suggests a sub-diffusive transport [25, 26].

We couple the system to a Markovian heat bath which does not affect the number of fermions in the system. Specifically, we assume that the density matrix of the system evolves via the Lindblad master equation [48],

$$\frac{\partial \hat{\rho}(t)}{\partial t} = -i \left[ \hat{H}, \hat{\rho}(t) \right] + \sum_i \left( \hat{L}_i \hat{\rho}(t) \hat{L}_i^{\dagger} - \frac{1}{2} \left\{ \hat{L}_i^{\dagger} \hat{L}_i, \hat{\rho}(t) \right\} \right), \tag{2}$$

where $\{.\}$ is the anti-commutator, the first term represents the unitary evolution and the second term gives the non-unitary dynamics. The operators $\hat{L}_i$ are Lindblad jump operators, which we take to be

$$\hat{L}_i = \sqrt{\gamma_i} \hat{n}_i, \tag{3}$$

where $\gamma_i$ represents the strength of the dissipation on site $i$. The dimensions of the density matrix are $\mathcal{N} \times \mathcal{N}$, where $\mathcal{N}$ is the Hilbert space dimension. This unfavorable scaling with the

system size makes the numerical solution of the Lindblad equation computationally expensive. A more efficient approach is to unravel the evolution into a *unitary* evolution of wavefunctions in the presence of white noise and then to average over the realizations of the noise to obtain the quantities of interest [35, 49–53].

For unitary unraveling, the time evolution operator is given by,

$$\hat{U}(t+dt,t) = e^{-i\hat{H}dt - i\sqrt{\gamma dt}\sum_i \eta_i(t)\hat{n}_i}, \tag{4}$$

where $\eta_i(t)$ are independent Gaussian random variables with mean zero and unit variance. The density matrix can be obtained by averaging over the realizations of the noise,

$$\hat{\rho}(t+dt) = \overline{|\psi(t+dt)\rangle\langle\psi(t+dt)|}, \tag{5}$$

where $|\psi(t+dt)\rangle = \hat{U}(t+dt,t)|\psi(t)\rangle$ and the over-bar represents averaging over the noise realizations. In this work, we set the noise strength to $\gamma = 1$ and the time-step to, $dt = 0.1$. We have corroborated that our results do not change if $dt$ is further reduced. We average over 10 noise trajectories and over 10 phase realizations ($\phi$ in Eq. (1), see Ref. [35] for further numerical details). We have found this averaging sufficient to reduce the statistical uncertainty of the data. To study the transport properties, we consider the following observables:

**Particle transport at infinite temperature.**    It is easy to check that irrespective of $\hat{H}$ the RHS of Eq. (2) vanishes for Hermitian Lindblad jump operators and $\hat{\rho} \propto \mathbb{I}$, such that any initial state will reach an infinite temperature state. Therefore, to avoid transient effects we directly characterize particle transport at infinite temperature. We note, that this corresponds to averaging over all possible particle sectors. For this purpose we look at the two-point density-density correlation function,

$$C_i(t) = \text{Tr}\left[\hat{\rho}_\infty\left(\hat{n}_i(t) - \frac{1}{2}\right)\left(\hat{n}_{L/2} - \frac{1}{2}\right)\right], \tag{6}$$

where $\hat{\rho}_\infty = \mathbb{I}/\mathcal{N}$ is the infinite-temperature density matrix, and $\hat{n}_{L/2}$ corresponds to an initial density excitation at the center of the lattice. From the Lindlbad master equation, the dynamics of such a correlation function can be obtained using a quantum regression theorem, which holds for any initial density matrix and allows the calculation of the correlation function in the stationary state of the Lindblad master equation [54]. A more efficient approach, that we employ here, is to evaluate the correlation function within the unitary unraveling dynamics. For non-interacting systems, Eq. (6) can be written as

$$C_i(t) = \left|\text{Tr}\left[\hat{\rho}_\infty \hat{a}_i^\dagger(t)\hat{a}_{L/2}\right]\right|^2 = \frac{1}{4}\left|U_{i,L/2}^s(t,0)\right|^2, \tag{7}$$

where $U_{i,L/2}^s(t,0)$ is the single-particle propagator, which can be efficiently calculated using Eq. (4),

$$U_{i,L/2}^s(t = N_t \cdot dt, 0) = \left\langle i\left|\prod_{n=1}^{N_t} e^{-i\left(h_s dt + \sqrt{\gamma dt}\eta_n\right)}\right|\frac{L}{2}\right\rangle. \tag{8}$$

Here $N_t = t/dt$ is the number of time-steps, $h_s$ is the single-particle Hamiltonian corresponding to Eq. (1) and the states $|i\rangle$ and $|L/2\rangle$, correspond to a particle found in sites $i$ and $L/2$ respectively. The second equality in Eq. (7) follows, since for unitary unraveling $\hat{a}_i^\dagger(t)$ can be written as: $\hat{a}_i^\dagger(t) = \sum_k U_{ik}^s(t,0)\hat{a}_k^\dagger$ and $\text{Tr}\left[\hat{\rho}_\infty \hat{a}_i^\dagger \hat{a}_j\right] = \frac{1}{2}\delta_{ij}$. Since $U_{i,L/2}^s(t,0)$ has dimensions $L \times L$, this allows us to efficiently study the system for large $L$.

We compute the root mean squared displacement (RMSD) [4, 6, 55, 56],

$$R(t) = \sqrt{\sum_{i=1}^{L}\left(i - \frac{L}{2}\right)^2 C_i(t)}, \tag{9}$$

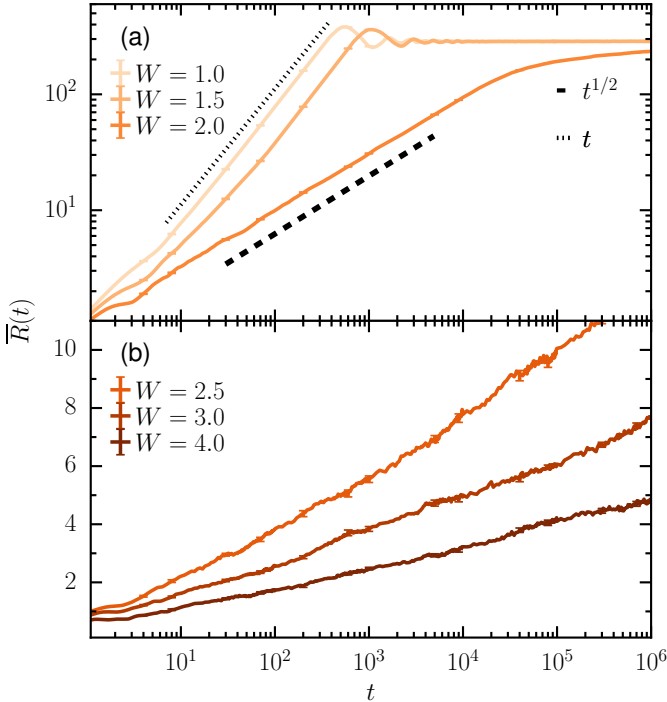

Figure 2: Root mean-squared displacement $\overline{R}(t)$ as a function of time; for system size $L = 1000$. The noise strength is set to $\gamma = 1$. The top panels, which are plotted on a log-log scale, correspond to the delocalized phase and the critical point. The bottom panels, which are plotted on a semi-log scale, correspond to the localized phase. The black dashed and dotted lines provide a guide to ballistic and diffusive transport, respectively. The statistical error is of the order of the line width.

for each trajectory of the noise and potential realization. We then average over different realizations to obtain the averaged RMSD, $\overline{R}(t)$. Typically, the RMSD grows as a power law in time, $\overline{R}(t) \sim t^{\alpha}$, where the dynamical exponent $\alpha = 1/2$ corresponds to diffusive transport and $\alpha = 1$ to systems with ballistic transport. Regimes characterized by sub-diffusive and super-diffusive transport involve exponents ranging between $0 \leq \alpha < 0.5$ and $0.5 < \alpha < 1$, respectively. For localized systems $\alpha = 0$.

## 3 Results

In this Section, we discuss our results for local coupling the heat bath to one site of the system and to a finite fraction of system sites. In finite fraction coupling we either couple the heat bath to a central region of the chain or to equally separated lattice sites through the entire chain (Fig.1). All the results below are obtained for a chain length of $L = 1000$.

### 3.1 Local coupling

In the left panels of Figure 2 we plot the dynamics of the root mean squared displacement for the delocalized, critical, and localized phases of the AAH model. In the absence of coupling to the heat bath, particle transport in these phases is ballistic, anomalous and absent, respectively [25, 26]. As can be seen from Fig. 2(a), local heat bath does not affect the nature of transport in the delocalized and critical regimes. On the other hand, it induces logarithmic transport

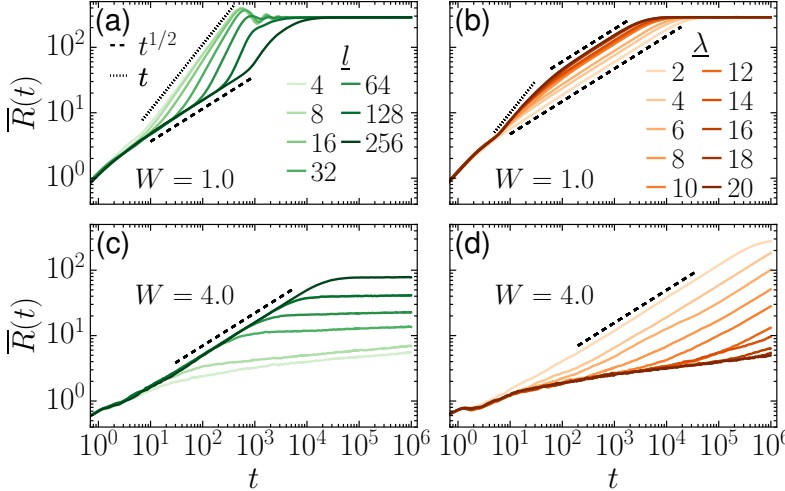

Figure 3: Root mean-squared displacement $\overline{R}(t)$, as a function of time for $L = 1000$. Left panels correspond to coupling of the heat bath to a central region of length $l$. Right panels to coupling the heat bath to system sites which are $\lambda$ distance apart. Top panel correspond to $W = 1$ and bottom panels to $W = 4$. Dashed and dotted lines are guides to the eye for diffusive and ballistic transport, respectively. More intense colors stand for larger $\lambda$ or $l$. The statistical error is of the order of the line width.

in the localized phase, similar to the case of the disordered Anderson model [35]. Within the time range considered, there is no sign of a crossover to diffusion, as opposed to the case where the noise is coupled to *all* the sites [34, 57–60].

## 3.2 Coupling to a finite part of the chain

We have seen that local coupling to the heat bath is not affecting transport in the delocalized phases and is inducing logarithmic transport in the localized phase. In this section we study how transport is affected when the heat bath is coupled to a finite fraction of system sites. Moreover, we will show that the spatial configuration of the coupled sites is important. Specifically we consider two different configurations: Coupling $l < L$ sites at the central region of the chain, or coupling sites which are separated a distance $\lambda$ apart, such that their density is $1/\lambda$.

The left panels of Fig. 3 show the dynamics of $\overline{R}(t)$ in the delocalized phase (top panel) and the localized phase (bottom panel) for different widths $l$ of the coupled central region. In both delocalized and localized phases $\overline{R}(t)$ initially grows diffusively. After this initial diffusive growth it crosses over to ballistic transport (Fig. 3(a)) in the delocalized phase, or to logarithmic transport in the localized phase (Fig. 3(c)). In both cases the crossover time, which we will designate by $t_l$, grows with the width of the coupled region, $l$.

The right panels of Fig. 3 show $\overline{R}(t)$ for coupling sites which are a distance $\lambda$ apart from each other. In the delocalized and localized phases, we observe a crossover to diffusion. At the critical phase, transport remains practically unaffected by the heat bath (not shown). In the delocalized phase, the initial transport is ballistic (see Fig. 3(b)) and in the localized phase, the initial transport is logarithmic (see Fig. 3(d)). In all cases the crossover time, which we denote by $t_d$ increases with $\lambda$. In the next section we introduce a classical master equation, which provides an explanation to the dependence of the crossover times on the parameters of the system.

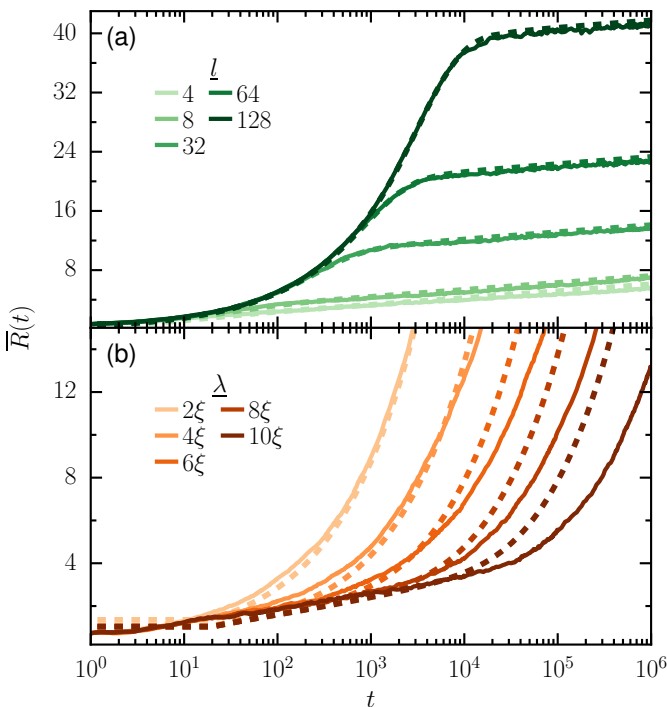

Figure 4: Root mean-squared displacement $\overline{R}(t)$, as a function of time; for a number of coupled central region widths $l$ (top panel) or the distance between the coupled sites, $\lambda$ (bottom panel). Solid lines correspond to the the numerical solution of the Lindblad equation (2), and dashed lines to the solution of the classical master equation (10), with time rescaled by a factor of 20 to have the best fit with the solid lines. The parameters used are $L = 1000$ and $W = 4$. The error-bars are of the order of the line width.

## 3.3 Classical picture

The dephasing mechanism of the heat bath, diminishes the importance of interference effects, and gives hope that classical treatment might be sufficient to understand the underlying phenomenology. We therefore follow the variable-range hopping approach of Mott [29], and assume that coupling the system to the heat bath induces transitions between localized single-particle eigenstates. The probability to find a particle in a single-particle state $\alpha$, which we denote by $p_\alpha$, evolves according to a classical master equation [57, 61, 62],

$$\partial_t p_\alpha = \sum_\beta \left( \Gamma_{\alpha\beta} p_\beta - \Gamma_{\beta\alpha} p_\alpha \right), \tag{10}$$

where the transition rates $\Gamma_{\alpha\beta}$ between the eigenstates $\alpha$ and $\beta$ can be calculated from first-order perturbation theory (see Appendix A),

$$\Gamma_{\alpha\beta} = \Gamma_{\beta\alpha} = \gamma \sum_{k \in \text{coupled sites}} \left| \phi_\beta^*(k) \phi_\alpha(k) \right|^2, \tag{11}$$

where $\gamma$ is the strength of the coupling, $\phi_\alpha(i)$ are the single particles states in the position basis, and the sum $k$ runs over the sites coupled to the heat bath. In the localized phase, by definition, $\phi_\alpha(i)$ decays exponentially. We order the single-particle eigenfunctions such that $\phi_\alpha(i)$ has it center of mass around a site $\alpha$, which allows us to approximate the transition

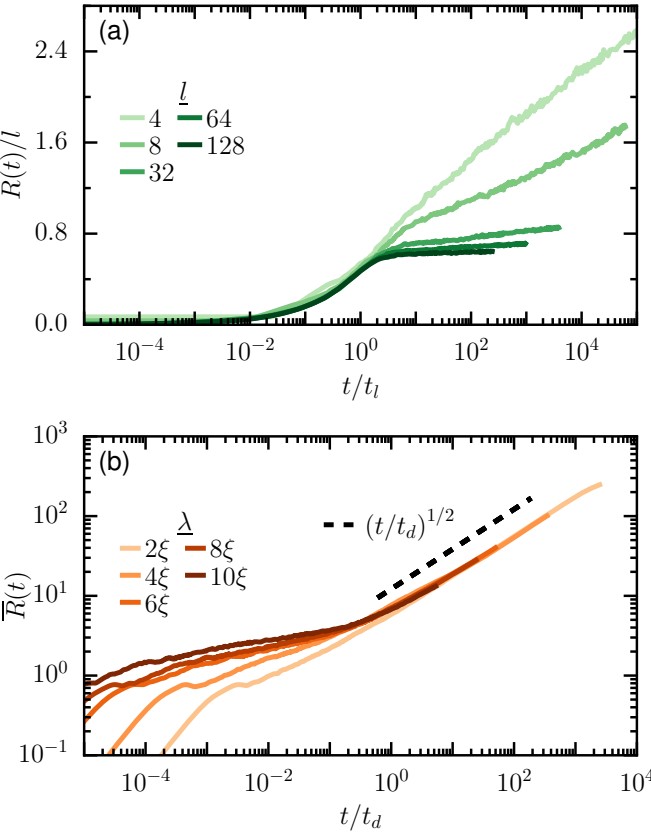

Figure 5: Same as Fig. 4 but with the time axis rescaled by the crossover times $t_l$ or $t_d$ (see main text).

rates as,

$$\Gamma_{\alpha\beta} = \gamma \sum_{k \in \text{coupled sites}} e^{-|\alpha-k|/\xi} e^{-|\beta-k|/\xi}, \tag{12}$$

where $\xi$ is the localization, which for the AAH model is $\xi = 1/\ln \frac{W}{2}$ [63]. The coupled site is, therefore, initiating transitions, or scattering, between localized states within its neighborhood. Transitions to far-lying states are exponentially suppressed with the distance from the coupled site. For local coupling, this leads to logarithmic transport, $\overline{R}(t) \sim \xi \ln(Jt)$ [35]. We note in passing that in the delocalized side, all the eigenstates $\phi_\alpha(k)$ are extended and therefore cannot be ordered by their center of mass, such that the spatial structure of the rates is lost. Moreover, the rates $\Gamma_{\alpha\beta}$ are of the same order of magnitude which means that transport is expected to transition to diffusion on a time-scale $\gamma^{-1}$. In what follows, we only focus on the localized phase.

When the heat bath is coupled to a region of final width $l$, the transitions rates $\Gamma_{\alpha\beta}$ are approximately constant in this region, and therefore a particle initiated in the coupled region is expected to diffuse. It takes the particle time, $t_l \sim l^2$, to leave this region. After leaving the coupled region, transitions rates exponentially decay with the distance from the region, and transport in the system is expected to be equivalent to the situation of local coupling. When the coupled sites are at equal distances $\lambda$ apart, on a time-scale of moving between two nearby coupled sites, transport is logarithmic, but at larger time-scales the expected motion is diffusive. The crossover time can be obtained from $\lambda = \xi \ln(Jt_d)$, yielding $t_d \sim J^{-1} \exp(\lambda/\xi)$, and the diffusion constant as $D \sim \lambda^2/t_d = \lambda^2 \exp(-\lambda/\xi)$.

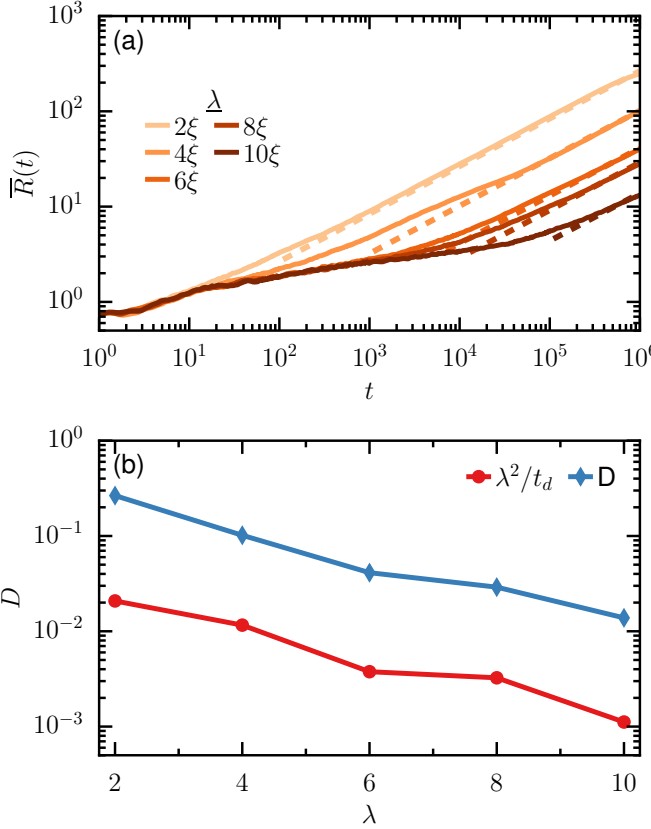

Figure 6: Extraction of the diffusion coefficient for various $\lambda$ by fitting $Dt^{1/2}$ to $\overline{R}(t)$ obtained from the solution of the Lindblad equation (dashed lines, top panel). The blue line at the bottom panel shows the diffusion coefficient as a function of $\lambda$ for $W = 4$ and $L = 1000$. The red line line corresponds to the theoretical prediction, $D \sim \lambda^2/t_d = \lambda^2 \exp(-\lambda/\xi)$, with $\xi = 1/\ln\frac{W}{2}$ (see text).

In Fig. 4 we compare the numerical solution of the classical master equation (10) to the numerical solution of the Lindblad equation (2). Since the classical rates (12) are obtained phenomenologically, the overall prefactor of the rates cannot be determined from microscopic considerations. We determine it by rescaling the time axis of the classical master equation such that the correspondence with the solution of the Lindblad equation is optimal (2). Apart from this trivial rescaling of the units of time, there are no fitting parameters. Remarkably, the agreement between the classical master equation and the Lindblad equation goes *beyond the qualitative level* for the central coupled region (top panel). For equal spacing coupling, there is still qualitative agreement, but the *quantitative* agreement is reasonable only for small $\lambda/\xi$.

In Fig. 5 we test the predictions of the classical theory for the crossover times, by rescaling of the time axis by $t_d$ or $t_l$, respectively. We see that such a rescaling correctly identifies the crossover time for both couplings to the heat bath when either $l$ or $\lambda$ are varied. The prediction for the diffusion constant is verified in Fig. 6. The agreement is not quantitative, however the exponential decrease of the diffusion constant with $\lambda$ is nicely captured.

# 4 Discussion

Using the unitary unraveling of the Lindblad master equation, we study the dynamical properties of the AAH model coupled to a dephasing heat bath. We consider local coupling and coupling to a finite part of the chain. This setup is partly motivated by dynamics in MBL systems in the presence of finite density of ergodic bubbles [7, 39, 40, 64], with the crucial difference that it is dissipative.

For local coupling of the heat bath in the delocalized and critical phases of the AAH model, we didn't observe any qualitative effects of the heat bath on the dynamics of the particle. On the other hand, in the localized phase, the root mean square displacement, entanglement entropy, and average energy (not shown) show asymptotic logarithmic growth, as it occurs for the one-dimensional Anderson insulator in the presence of a local noise [35]. Suppose the region of the coupling to the bath is of finite width. In that case, we find a regime of transient diffusion, which crosses over to ballistic transport in the delocalized phase and logarithmic transport in the localized phase. We have shown that this crossover time increases as the square of the width of the region, as typical for diffusion. When the heat bath is coupled to the system on equally spaced sites, initial transport in the system is similar to the transport with local coupling. Eventually, it crosses over to diffusion in all phases. Specifically, in the localized phase, the crossover time increases exponentially with the distance between the coupled sites.

We have shown that in the localized phase, a classical master equation with transition rates that exponentially decay from the location of the coupled sites captures all the observed phenomenology. Moreover, it provides accurate predictions of crossover times for all studied couplings of the heat bath. This time scale is set by the time it takes for a particle to traverse the distance $\lambda$ between two nearby coupled sites. In the delocalized phase, this time is proportional to $\lambda$ (if the transport is ballistic) or $\lambda^2$ (if it is diffusive). On the other hand, it scales as $t_d \propto \exp(\lambda/\xi)$ in the localized phase. Within the classical model, the motion of the particle between the coupled sites can be viewed as a random walk with a spatial step of $\lambda$ and a mean-free time of $t_d$, which means that the motion is diffusive, with diffusion constant given by $D \sim \lambda^2/t_d = \lambda^2 \exp(-\lambda/\xi)$.

In this work, we have focused only on fixed $\lambda$, such that $D$ is also fixed. If $\lambda$ is allowed to vary randomly, such that its distribution is unbounded, then the average time to transition between coupled sites, $t_d$, can diverge. In this case, the average diffusion coefficient will vanish, and transport will be subdiffusive (see Refs. [36, 37] and Appendix B). Furthermore, this finite number of dephasing sites in the non-interacting localized system mimics the presence of ergodic bubbles in the many-body localized system that acts as a *finite* thermal bath and induces quantum transport. However, unlike the ergodic bubbles, which cool off during the dynamics, and therefore can stop being ergodic, the system here does not affect the heat bath. As such, our model provides a non-rigorous upper bound on ergodic bubbles-induced transport in a many-body localized system [7, 39, 40, 64].

Here, we focus on a quasi-periodic model without mobility edges. In the presence of mobility edges, at least in principle, the coupling to the heat bath can create transitions between the localized and delocalized states. Several interesting questions arise for these models: would the coupling eliminate the intermittent logarithmic transport regime? How will the dynamics depend on the initial conditions? We leave these questions to future studies.

## Acknowledgments

**Funding information** This research was supported by a grant from the United States-Israel Binational Foundation (BSF, Grant No. 2019644), Jerusalem, Israel, and the United States Na-

tional Science Foundation (NSF, Grant No. DMR−1936006), and by the Israel Science Foundation (grants No. 527/19 and 218/19). D.S.B acknowledges funding from the Kreitman fellowship.

## A  Transition rates for white noise

In this section, we calculate the transition rates using the first order perturbation theory. Withing the unitary unraveling of the Lindblad equation the dynamics of the system is described by a the time-dependent Hamiltonian,

$$\hat{H}(t) = \hat{H}_0 + \hat{V}(t),$$

where, $\hat{V}(t) = \sqrt{\gamma} \sum_i \eta_i(t) |i\rangle\langle i|$ with $\eta_i(t)$ being a Gaussian random variable with zero mean and unit variance. The noise is characterized by the correlation function, $\langle \eta_i(t)\eta_j(t') \rangle = \delta_{ij}\delta(t-t')$. The time evolution of the state $|\psi(t)\rangle$ in the eigenbasis $|\alpha\rangle$ of $\hat{H}_0$ can be written as,

$$\partial_t c_\alpha = -i \sum_\gamma \langle \alpha | \hat{V}(t) | \gamma \rangle c_\gamma(t) e^{-i(\epsilon_\gamma - \epsilon_\alpha)t}, \tag{A.1}$$

where $\epsilon_\alpha$ is the eigenvalue of $\hat{H}_0$ which corresponds to $|\alpha\rangle$. In the integral form, this corresponds to,

$$c_\alpha(t) = c_\alpha(0) - i \int_0^t \sum_\gamma dt' \langle \alpha | \hat{V}(t) | \gamma \rangle c_\gamma(t) e^{-i(\epsilon_\gamma - \epsilon_\alpha)t'}. \tag{A.2}$$

Back substitution of the LHS into the RHS and ignoring terms of order, $O(V^2)$ gives,

$$c_\alpha(t) = c_\alpha(0) - i \int_0^t \sum_\gamma dt' \langle \alpha | \hat{V}(t) | \gamma \rangle c_\gamma(0) e^{-i(\epsilon_\gamma - \epsilon_\alpha)t'}. \tag{A.3}$$

We assume that the system is initialized in state $\beta$, such that $c_\gamma(0) = \delta_{\gamma\beta}$, which yields,

$$c_\alpha(t) = \delta_{\alpha\beta} - i \int_0^t dt' \langle \alpha | \hat{V}(t) | \beta \rangle e^{-i(\epsilon_\beta - \epsilon_\alpha)t'}. \tag{A.4}$$

The transition probability rate between state $|\alpha\rangle$ and $|\beta\rangle$ is given by

$$\Gamma_{\alpha\beta} = |c_\alpha(t)|^2 / t, = \frac{1}{t} \int_0^t dt' \int_0^t dt'' \langle \alpha | \hat{V}(t') | \beta \rangle \langle \beta | \hat{V}(t'') | \alpha \rangle e^{-i(\epsilon_\beta - \epsilon_\alpha)(t' - t'')}. \tag{A.5}$$

Now since averaging over the noise gives,

$$\overline{\langle \alpha | \hat{V}(t') | \beta \rangle \langle \beta | \hat{V}(t'') | \alpha \rangle} = \gamma \sum_{i,j} \overline{\eta_i(t')\eta_j(t'')} \langle \alpha | i \rangle \langle i | \beta \rangle \langle \alpha | j \rangle \langle j | \beta \rangle$$

$$= \gamma \sum_i |\langle \alpha | i \rangle|^2 |\langle \beta | i \rangle|^2 \delta(t' - t''), \tag{A.6}$$

we obtain

$$\Gamma_{\alpha\beta} = \frac{\gamma}{t} \sum_i |\langle \alpha | i \rangle|^2 |\langle \beta | i \rangle|^2 \int_0^t dt' \int_0^t dt'' \delta(t' - t'') e^{-i(\epsilon_\beta - \epsilon_\alpha)(t' - t'')}$$

$$= \gamma \sum_i |\langle \alpha | i \rangle|^2 |\langle \beta | i \rangle|^2,$$

where $\langle \beta | i \rangle = \phi_\beta^*(i)$ and $\langle k | \alpha \rangle = \phi_\alpha(k)$.

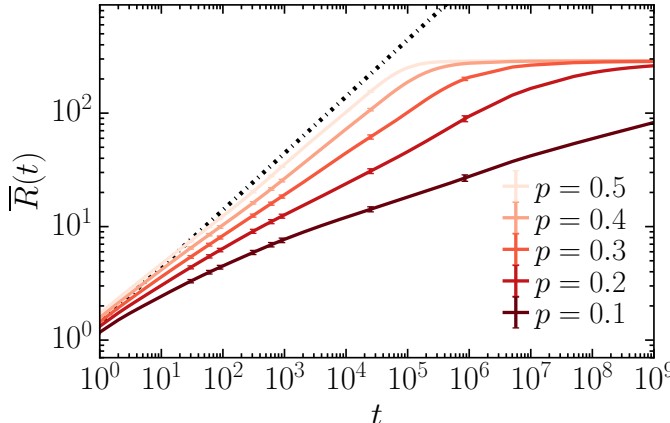

Figure 7: Dynamics of $\overline{R}(t)$ for for $W = 4.0$, and the case where each site of the system can be coupled with probability $p$. The dotted line corresponds to a guide to the diffusive transport $(t^{1/2})$.

## B   Coupling at random sites

In this section, we consider the case where a given site is coupled with probability $p$, which means that the distance between two coupled sites is distributed according to the Poisson distribution. In Fig. 7 we shows the dynamics of $\overline{R}(t)$ calculated using the classical master equation for $W = 4.0$ and and number of $p$. We average the data over 50 realizations of the couplings to the heat bath. As opposed to the case where the coupled sites are placed at an equal distance from each other leading to eventual diffusion, here we find sub-diffusive transport with a dynamical exponent which depends on $p$. Our results are consistent with Refs. [36,37] where the stationary state current is calculated in a similar system.

## C   Dynamics of entanglement entropy

In this section we study the dynamics of entanglement entropy for a single site coupling to the bath. While the entanglement entropy is not a good measure of quantum information for mixed states [65], for the unitary unraveling, it is well defined for each of the trajectories. Therefore, in situations when one can physically justify the specific form of the unraveling,[1] the entanglement averaged over the various trajectories is a sensible quantity [35,66]. Another advantage of the unitary unraveling is that the wavefunction $|\psi(t)\rangle$, is Gaussian through the entire evolution, which allows an efficient computation of the entanglement entropy, using the relation,

$$S(t) = -\sum_{\alpha} [c_{\alpha}(t) \ln c_{\alpha}(t) + (1 - c_{\alpha}(t)) \ln(1 - c_{\alpha}(t))] . \tag{C.1}$$

Here, $c_{\alpha}(t)$ are eigenvalues of the correlation function $\langle \psi(t) | \hat{a}_i^{\dagger} \hat{a}_j | \psi(t) \rangle$, restricted to the subsystem of interest [67,68]. For ballistic transport, the entanglement entropy grows linearly with time, while anomalous transport is characterized by a sub-linear growth [69–72].

In the right panels of Fig. 8 we present the growth of the entanglement entropy starting from a random product state averaged over noise trajectories ($\overline{S}(t)$). We use a random product state so that initially the entanglement entropy is zero, which provides us with a finite regime of entanglement entropy growth. In particular, we initiate the system at a random product state,

---

[1]For example, a system in a time-dependent noisy potential.

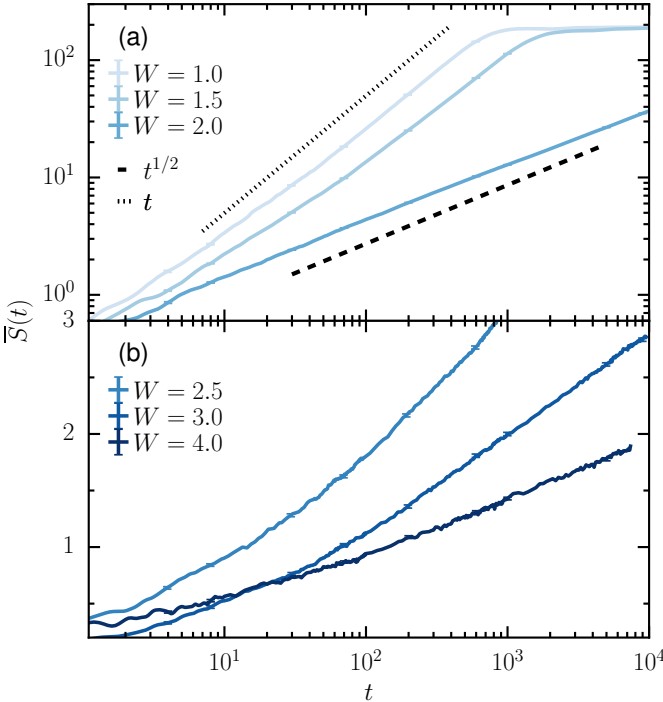

Figure 8: Entanglement entropy $\overline{S}(t)$ as a function of time; for system size $L = 1000$. The noise strength to $\gamma = 1$. The top panels, which are plotted on a log-log scale, correspond to the delocalized phase and the critical point. The bottom panels, which are plotted on a semi-log scale, correspond to the localized phase. The statistical error is of the order of the line width.

namely, $\rho^s_{ij}(t = 0) = n_j \delta_{ij}$, with random $n_j \in 0, 1$. In the delocalized regime, the growth of the entanglement entropy is unaffected by the presence of the local heat bath and is consistent with a power-law dependence on time, observed in closed systems [63]. In the localized case, the heat bath induces logarithmic growth of entanglement entropy which is reminiscent of the entanglement entropy growth in the MBL phase [73–76]. However, here this growth is accompanied with logarithmic particle transport, while the presence of logarithmic particle transport in the MBL phase is under debate [77–79].

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
