# Peer review of "Noise-induced transport in the Aubry-André-Harper model"

_SciPost Physics, doi:SciPost Phys. Core 7, 023 (2024)_

## Round 1 · Referee Report · Anonymous (Referee 1) · 2023-8-27

Strengths

1)Numerical results are compared to phenomenological classical master equation, showing excellent agreement
2)Qualitative interpretation of the results clearly explained and discussed

Weaknesses

1)Incremental with respect to previous literature, lack of major advance/breakthrough
2)Introduction and motivations can be improved
3)Results on entanglement entropy only for one type of system-bath coupling

Report

This work studies transport in the Aubry-Andre'-Harper (AAH) model in presence of a Markovian dephasing bath. Different types of coupling geometries are considered. The focus is on spreading of density excitations (encoded in the root mean squared displacement) and partially on entanglement entropy.

The numerical results, obtained through a unitary unravelling of the Lindblad master equation, are well presented and interpreted through a classical master equation. The Authors find that when the coupling to the bath affects a finite portion of the chain transport at short times becomes diffusive, independent on disorder, while at longer time scales (outside of the zone of the influence of the bath) the usual transport behavior of the AAH is restored.
On the other hand, when the coupling to the bath is distributed among equally spaced sites diffusion emerges at long times.

I find this work interesting for those working in the field of disorder and dynamics. However I do not see the case for publication in Scipost Physics, based on its acceptance criteria. In particular I do not see a major result/breakthrough here. The work appears to be incremental with respect to previous literature (in particular Ref 58 from a subset of the authors, where similar analysis was done for the one dimensional Anderson model in presence of a local bath). I think the manuscript could be published in SciPostCore, provided the Authors address the points below.

Requested changes

1)Introduction and general motivations could be improved: why precisely the Authors are considering this type of coupling to the bath? What is the connection with MBL?Shouldn't thermal inclusions be somehow randomly distributed ?
2)Related question: in the conclusions the Authors mention their results provide an upper bound on transport in MBL. Why? How this comes about? Can the Authors elaborate on this point?
3)While the unravelling of the density matrix in Eq.5 is quite clear, it's less obvious how one should interpret Eq. 6, particularly given the Authors consider directly an infinite temperature density matrix. Is this correlation function averaged over the noise? Should this be equivalent to computing the same object through Linbdlad master equation because of some quantum regression theorem? It seems that some clarification is needed here.
4)Results on entanglement entropy are presented only in Fig 2, for the local (single-site) bath coupling? Why? As it is, this part seems quite decoupled from the rest of the article.
5)I am a bit surprised that a time-step dt=0.1 (in units of the hopping I guess?) is enough to have converged data and even more that averaging over N=10 trajectories is enough to converge to the Lindblad result. Could the Authors provide some supporting evidence?

  • validity: high
  • significance: good
  • originality: ok
  • clarity: high
  • formatting: perfect
  • grammar: perfect

Author:  Devendra Singh Bhakuni  on 2023-11-06  [id 4096]

(in reply to Report 1 on 2023-08-27)

We thank the Referee for their thorough review of the manuscript and recommendation for publication in SciPost Physics Core. It is important to emphasize that Ref.[58] was focused on a different question: How does a local coupling to a heat bath affect transport in a localized model? In contrast, our current work presents a comprehensive analysis of various types of couplings for the bath, motivated by both theoretical and experimental questions. Moreover, it provides a broader theoretical picture, which we verify using an experimentally relevant model which exhibits different transport regimes. We show that using the configuration of the couplings, transport can be tuned to logarithmic, sub-diffusive, or diffusive. We provide a qualitatively good estimate of the diffusion coefficient in the diffusive regime.

Besides the broad implication on transport in open quantum systems, our study also captures the physics of ergodic bubbles in the many-body localized phase. It sheds light on the nature of transport in this phase. It is worth mentioning that similar setups to study the effect of ergodic bubbles have been previously proposed theoretically (Phys. Rev. Lett. 119, 150602) and experimentally (Nature Physics 19, 481–485 (2023)) but with no consideration of quantum transport.

Given the above, our study goes beyond Ref. [58] and is directly relevant to the fields of quantum transport in the presence of dephasing, Anderson localization, and many-body localization. Moreover, it has direct theoretical and experimental applications. As such, we believe it matches the criteria of SciPost Physics.

The referee writes:

Introduction and general motivations could be improved: why precisely the Authors are considering this type of coupling to the bath? What is the connection with MBL? Shouldn't thermal inclusions be somehow randomly distributed?

Our response: The main goal of our work was to consider the effect of the spatial configuration of the coupling to the heat bath on quantum transport, which is a question of great interest in the field of open quantum systems. As such, we considered types of couplings relevant to several experimental and theoretical settings. Local coupling can be relevant to systems of point contacts (Phys. Rev. Lett. 92, 156801, 2004, Phys. Rev. B 85, 155327 (2012)) and the thermalization of non-ergodic chains by a single ergodic bubble (see, ((Phys. Rev. Lett. 119, 150602), for theoretical study and Nature Physics 19, 481–485 (2023) for experimental)). Periodic coupling is relevant, for example, when a system is coupled to an underlying periodic lattice/crystal (Phys. Rev. Lett. 125, 180605). In the context of many-body localization, the ergodic bubbles can be modeled by coupling to an external bath. The positions of the ergodic bubbles are indeed random for disordered models (see the Appendix of our work for results on randomly distributed couplings). However, for the Aubry-Andre model that we consider in this work, the location of the ergodic bubbles is deterministic, highlighting the importance of studying other configurations.

We have considerably reworked the introduction, providing a more general motivation for our work.

The referee writes:

Related question: in the conclusions the Authors mention their results provide an upper bound on transport in MBL. Why? How this comes about? Can the Authors elaborate on this point?

Our response: In the many-body localized phase, the ergodic bubbles serve as a \textit{finite} thermal bath and induce quantum transport. We model those rare regions/ergodic bubbles by an \textit{infinite} dephasing bath and consider the simplest case of a non-interacting localized system. Unlike the ergodic bubbles, which cool off during the dynamics (and therefore can stop being ergodic), the system does not affect the heat bath. As such, our model provides a (non-rigorous) upper bound on ergodic bubbles-induced transport in a many-body localized system. We have clarified this point in the text.

The referee writes:

While the unravelling of the density matrix in Eq.$5$ is quite clear, it's less obvious how one should interpret Eq.$6$, particularly given the Authors consider directly an infinite temperature density matrix. Is this correlation function averaged over the noise? Should this be equivalent to computing the same object through Linbdlad master equation because of some quantum regression theorem? It seems that some clarification is needed here.

Our response: The referee is right that for a system coupled to a bath, the evolution of the correlation function is obtained using the quantum regression theorem. This allows us to compute the correlation function directly in the stationary state of the Lindblad equation, which for Hermitian jump operators is the infinite temperature. For the problem at hand, it is much more efficient to use the unraveling of the dynamics, which is unitary and allows computation of the correlation function in a standard fashion, as explained after Eq.~(6). We have added the appropriate clarification in the text.

The referee writes:

Results on entanglement entropy are presented only in Fig $2$, for the local (single-site) bath coupling? Why? As it is, this part seems quite decoupled from the rest of the article.

Our response: We agree that this part looks decoupled from the rest of the article. Since we are primarily focused on the quantum transport, we have moved the analysis of the entanglement entropy from the main text to the Appendix.

The referee writes:

I am a bit surprised that a time-step $dt=0.1$ (in units of the hopping I guess?) is enough to have converged data and even more that averaging over $N=10$ trajectories is enough to converge to the Lindblad result. Could the Authors provide some supporting evidence?

Our response: The unitary unraveling used in the work is exact for any $dt$ for Hermitian Lindblad jump operators. Increasing the number of trajectories suppresses the statistical noise, and for our purpose, averaging over $10$ trajectories along with $10$ different choices of the on-site potential was enough to suppress the statistical noise. A relationship between the unraveling and the Lindblad master for any choice of $dt$ is provided in Phys. Rev. Lett. 118,140403 (2017). For completeness, we plot (see attached file) the root mean squared displacement dynamics for the case where the bath is coupled to all the sites. It can be seen that the qualitative features of the dynamics are independent of the choice of $dt$.

Attachment:

width_dt_L100_oLZYrd8.pdf

---

## Round 1 · Referee Report · Anonymous (Referee 2) · 2023-9-6

Report

This paper seems to have solid and original results. They are not ground-breaking, but will be of interest to some other researchers working in this general area (such as myself).

I suggest a few changes to help with the clarity and completeness of the presentation:

The classical approximations that are developed in section IIIC appear to be only for the localized phase, which is fine. But this restriction should be explicitly stated where this is mentioned in the abstract, discussion and in section IIIC, since other parts of the paper also treat the critical and ballistic phases.

The study of the entanglement entropy uses an initial random product state. This initial state should be described more completely. Since another part of the paper uses an initial mixed state, here it should be said explicitly that an initial pure state is used. The ensemble from which this random pure product state is drawn should be explicitly stated. I suspect they mean initial local eigenstates of the occupation, randomly occupied or empty with equal probabilities for these two possibilities, but this must be explicit. Or are the initial local pure states uniformly sampled from the local Bloch sphere?

The caption of Fig. 2 should say what bath and couplings are being used (I did not find those details in the main text either).

Fig. 6(b), especially the lower trace, shows a possible odd/even effect vs lambda/2. Or is this all within the error bars? No error bars are shown in any of the figures. They should be roughly estimated and indicated, or where they are not visible this should be stated. This is crucial and required for any numerical work where the errors can be estimated (as they can here).
  • validity: high
  • significance: good
  • originality: good
  • clarity: good
  • formatting: -
  • grammar: -

Author:  Devendra Singh Bhakuni  on 2023-11-06  [id 4097]

(in reply to Report 2 on 2023-09-06)

We thank the Referee for the positive comments and a constructive report. We have incorporated all of the suggestions.

The referee writes:

The classical approximations that are developed in section IIIC appear to be only for the localized phase, which is fine. But this restriction should be explicitly stated where this is mentioned in the abstract, discussion and in section IIIC, since other parts of the paper also treat the critical and ballistic phases.

Our response: As shown in Appendix A, the classical model applies to any system driven by white noise. However, for delocalized models, by definition, the states $\phi_{\alpha}(k)$ are not ordered by the "center-of-mass" of the states, such that the spatial structure of the rates is lost. Moreover, the rates $\Gamma_{\alpha\beta}$ are all roughly equal. Therefore, for such models, transport is expected to transition to diffusion on time-scale $\gamma^{-1}$. We have added a clarifying statement in Section IIIC.

The referee writes:

The study of the entanglement entropy uses an initial random product state. This initial state should be described more completely. Since another part of the paper uses an initial mixed state, here it should be said explicitly that an initial pure state is used. The ensemble from which this random pure product state is drawn should be explicitly stated. I suspect they mean initial local eigenstates of the occupation, randomly occupied or empty with equal probabilities for these two possibilities, but this must be explicit. Or are the initial local pure states uniformly sampled from the local Bloch sphere?

Our response: We have added a more detailed description of the initial state.

The referee writes:

The caption of Fig.$2$ should say what bath and couplings are being used (I did not find those details in the main text either).

Our response: We consider $\gamma=1$. We have now added this in the revised manuscript.

The referee writes:

Fig. 6(b), especially the lower trace, shows a possible odd/even effect vs $\lambda/2$. Or is this all within the error bars? No error bars are shown in any of the figures. They should be roughly estimated and indicated, or where they are not visible this should be stated. This is crucial and required for any numerical work where the errors can be estimated (as they can here).

Our response: We are unsure about the even/odd effect. The error bars in our simulations are less than the line width. We have now mentioned this.

---

## Round 2 · Referee Report · Anonymous (Referee 2) · 2023-11-8

Report

Yes, the authors have made appropriate improvements. Except:

The last sentence of the caption to Fig 6 says: "The dashed blue line corresponds to the theoretical prediction, with no fitting parameters." But there is no dashed blue line in the figure. And I could not find a theoretical prediction without fitting parameters for D anywhere in the text. So this needs to be fixed in multiple ways. Make it very clear what theoretical prediction you are showing, and why (how) such a prediction can be made without any fitting parameters (and, if I understand, end up off by an order of magnitude or so). My guess is that there really is not any such prediction, just an expected scaling with an unknown multiplicative factor (which thus ends up as a fitting parameter).
  • validity: -
  • significance: -
  • originality: -
  • clarity: -
  • formatting: -
  • grammar: -

Author:  Yevgeny Bar Lev  on 2023-11-28  [id 4152]

(in reply to Report 1 on 2023-11-08)

**The referee writes:**
>The last sentence of the caption to Fig 6 says: "The dashed blue line corresponds to the theoretical prediction, with no fitting parameters." But there is no dashed blue line in the figure. And I could not find a theoretical prediction without fitting parameters for D anywhere in the text. So this needs to be fixed in multiple ways. Make it very clear what theoretical prediction you are showing, and why (how) such a prediction can be made without any fitting parameters (and, if I understand, end up off by an order of magnitude or so). My guess is that there really is not any such prediction, just an expected scaling with an unknown multiplicative factor (which thus ends up as a fitting parameter)

**Our response:**
We thank the referee for finding the problem in the caption. We have fixed it and added the explicit expression for the theoretical prediction. The referee is correct that the theoretical prediction is based on the analysis of the relevant time and length scales and, therefore, cannot be expected to match the numerical value of D. Nevertheless, it does capture nicely its dependence on $\lambda$, up to a numerical prefactor.

---

## Round 2 · Referee Report · Anonymous (Referee 1) · 2023-11-12

Report

The Authors have improved the Introduction and provided a more clear motivation for their study. I still find the choice of bath coupling schemes rather artificial and therefore of narrow interest.
The results on entanglement entropy could have been expanded, instead the Authors decided to move them to the Appendix.

More importantly, I am still not fully satisfied with the Authors discussion of their theoretical approach: the single particle propagator above Eq. (7) is not defined. How are they solving the dynamics?What is the filling of the system? In section II they mention N particles in L sites, but I could not find the value of N. Are the Authors considering the one particle sector, N=1?
Finally, it is not obvious to me that in presence of a non-uniform bath coupling (for example scheme a) the stationary state is infinite temperature, as the Authors assume in this work. I could imagine some interesting crossover with \ell (the size of the region coupled to the bath), where the rest of the system act as another bath leading to a non-equilibrium steady state, with effective temperature ultimately going to infinity with \ell? The Authors should at least comment on this point.

Based on these comments I conclude the paper does not meet the criteria for SciPost Physics. If the Authors address the points above (clarification of their method and of the validity of the infinite temperature state) the manuscript can be accepted in SciPost Physics Core.
  • validity: -
  • significance: -
  • originality: -
  • clarity: -
  • formatting: -
  • grammar: -

Author:  Yevgeny Bar Lev  on 2023-11-28  [id 4151]

(in reply to Report 2 on 2023-11-12)

The referee writes:

The Authors have improved the Introduction and provided a more clear motivation for their study. I still find the choice of bath coupling schemes rather artificial and therefore of narrow interest.

Our response: In our previous response, we have explained that the theoretical framework is general and applies to all possible couplings. Therefore, we are surprised by the referee's comment that our results are narrowly interesting. Since we can only study specific choices numerically, we have decided to focus on the deterministic spacing of the couplings, which is more appropriate to the Aubry-Andre model and more experimentally relevant. Appendix B includes results for coupling to random sites.

The referee writes:

The results on entanglement entropy could have been expanded, instead the Authors decided to move them to the Appendix.

Our response: We accepted the comment in the initial report that the entanglement part is decoupled from the rest of the story. We moved this part to the Appendix to make the story more focused.

The referee writes:

More importantly, I am still not fully satisfied with the Authors discussion of their theoretical approach: the single particle propagator above Eq. (7) is not defined. How are they solving the dynamics?What is the filling of the system? In section II they mention N particles in L sites, but I could not find the value of N. Are the Authors considering the one particle sector, N=1?

Our response: We thank the referee for pointing this out. We have added a definition of the single-particle propagator, which follows directly from Eq. (4). The correlation function is calculated in the infinite temperature state, which averages over all particle sectors. We have added a clarifying comment above Eq (7) and removed N from the leading sentence of Sec II, which is not used anyway. We also added the average over noise trajectories (denoted by an overbar) which was missing in the definition of the root mean squared displacement and in the legend of the figure of the entanglement entropy in the appendix, which is of course averaged over noise trajectories as well.

The referee writes:

Finally, it is not obvious to me that in presence of a non-uniform bath coupling (for example scheme a) the stationary state is infinite temperature, as the Authors assume in this work. I could imagine some interesting crossover with \ell (the size of the region coupled to the bath), where the rest of the system act as another bath leading to a non-equilibrium steady state, with effective temperature ultimately going to infinity with \ell? The Authors should at least comment on this point.

Our response: Just above Eq (6), we write "It is easy to check that irrespective of H the RHS of Eq. (2) vanishes for Hermitian Lindblad jump operators and $\hat{\rho}\propto\mathbb{I}$", where Eq (2) is the Lindblad equation. Since this can be checked by a simple substitution, we decided not to add an explicit derivation in the text. Therefore, if the stationary state is unique, it must be the infinite temperature state, even when the coupling is only to one site.

---

## Round 3 · Referee Report · Anonymous (Referee 1) · 2024-2-28

Report

Dear Editor,

as I have mentioned in my previous reports I do not see a case for this manuscript to be published in SciPost Physics based on its acceptance criteria.
In particular I do not see a major advance/breakthrough here, or any substantial or major novelty with respect to previous literature (also from the same group, see Ref 35).
This is nevertheless a solid work which will certainly be of interest for the researchers working on localisation and its stability. The Authors have addressed my previous comments and therefore I am happy to recommend publication in SciPost Core.

---

## Editorial Decision

published